# Large Animal Models for Investigating Cell Therapies of Stress Urinary Incontinence

**DOI:** 10.3390/ijms22116092

**Published:** 2021-06-05

**Authors:** Bastian Amend, Niklas Harland, Jasmin Knoll, Arnulf Stenzl, Wilhelm K. Aicher

**Affiliations:** 1Department of Urology, University of Tuebingen Hospital, 72076 Tuebingen, Germany; bastian.amend@med.uni-tuebingen.de (B.A.); Niklas.Harland@med.uni-tuebingen.de (N.H.); urologie@med.uni-tuebingen.de (A.S.); 2Center of Medical Research, Department of Urology at UKT, Eberhard-Karls-University, 72076 Tuebingen, Germany; Jasmin.Knoll@med.uni-tuebingen.de

**Keywords:** urinary incontinence, disease models, animal, cell therapy

## Abstract

Stress urinary incontinence (SUI) is a significant health concern for patients affected, impacting their quality of life severely. To investigate mechanisms contributing to SUI different animal models were developed. Incontinence was induced under defined conditions to explore the pathomechanisms involved, spontaneous recovery, or efficacy of therapies over time. The animal models were coined to mimic known SUI risk factors such as childbirth or surgical injury. However, animal models neither reflect the human situation completely nor the multiple mechanisms that ultimately contribute to the pathogenesis of SUI. In the past, most SUI animal studies took advantage of rodents or rabbits. Recent models present for instance transgenic rats developing severe obesity, to investigate metabolic interrelations between the disorder and incontinence. Using recombinant gene technologies, such as transgenic, gene knock-out or CRISPR-Cas animals may narrow the gap between the model and the clinical situation of patients. However, to investigate surgical regimens or cell therapies to improve or even cure SUI, large animal models such as pig, goat, dog and others provide several advantages. Among them, standard surgical instruments can be employed for minimally invasive transurethral diagnoses and therapies. We, therefore, focus in this review on large animal models of SUI.

## 1. Introduction

Stress urinary incontinence is the most common form of incontinence. Despite intensive research over the last decades, causal treatments suitable for a substantial part of the patients affected are not available. Therefore, preclinical studies utilizing animal models deepen our knowledge on pathomechanisms contributing to the development of urinary incontinence, including especially SUI [1,2]. A general draw-back of most animal models is the fact that the anatomy of the lower pelvic floor of quadruped animals differs significantly from the anatomy of primates [3,4,5,6]. In addition, in quadruped animals the abdominal mass does not directly burden the pelvic floor, thus pressure on the urinary bladder is lower compared to bipedal species. However, ethical concerns limit studies with primates exploring the etiology of SUI, its diagnosis and therapy. However, a few detailed analyses provided evidence that rhesus monkeys share some important similarities in anatomy and physiology of the urethral sphincter muscles with higher primates, including male and female specificities [7]. Therefore, rhesus monkeys were included as animal models in a few specific SUI studies [8,9,10,11]. In these studies, the efficacy of SUI therapy by injection of muscle precursor cells was corroborated. In addition, functional sphincter recovery after application of chemokines was shown as well [11]. The studies confirmed earlier research indicating that not only cells differentiating possibly in muscle tissues but also bioactive components such as cytokines, activating intrinsic regenerative mechanisms, may contribute to SUI amelioration [12]. Activation of urethral satellite cells plays a key role in cytokine-driven sphincter repair [12].

However, rhesus monkeys inherit some specific disadvantages in the context of SUI research as well. Besides ethical issues, the efforts for breeding and husbandry of such animals are a concern. Therefore, most studies published on urinary incontinence utilized rodents [1,2,13,14]. The public attention to animal studies with rodents and raising awareness of their possible clinical benefit can be conveyed easier when compared to studies with monkeys or animals of affection, such as dogs or cats. To address this, several large animal studies were performed with “agricultural livestock”, sheep, goats and pigs. Again, these species offer advantages and disadvantages. Breeding and husbandry of farm animals are rather simple and well established. Researchers therefore can take advantage of herds of animals which translates into large cohorts, distinct regimens and robust statistics. Another advantage of such large species is the fact that standard surgical instruments can be used for most interventions. Large animals facilitate transurethral minimally invasive surgery instead of laparoscopic interventions or open surgery as required for studies in rodents, rabbits or other small animals. However, in contrast to rodents, knowledge and tools for analysis of blood and tissue samples such as specific antibodies, information on gene expression or genes themselves is not established in most institutes dealing with urological research. When designing an animal study to investigate pathomechanisms, diagnosis or therapy of urinary incontinence, these and many other aspects must be taken into consideration to yield a balanced and resilient project.

In this review, we will summarize current knowledge on well-established methods to induce experimental incontinence in animals, on determination of sphincter insufficiency and on pre-clinical studies to explore different regimens to ameliorate the sequela of malfunction of the urethral sphincter complex.

## 2. Large Animal Models to Study Urinary Incontinence

### 2.1. Different Methods to Establish Urinary Incontinence in Animals

Mechanical load and stretching of the lower pelvic floor as caused by pregnancy and vaginal delivery are known to increase the risk of developing SUI significantly. Therefore, early studies were designed to mimic injuries to the sphincter apparatus of the urethra by vaginal distension in female rats [13,15]. To this end a balloon catheter is introduced in the vagina of female rats (in deep anesthesia), inflated with variable amounts of fluid and slowly retracted. In this model, urethral sphincter injury included damage of the urethral smooth muscle and neuronal tissue [1,15]. Vaginal distension studies were performed in rats with quite different protocols: the balloon dilatation diameter (2 mL to 5 mL filling), dilatation time (up to 4 h), age/weight, or breeding status (virgin rats versus retired breeders) of the female animal employed varied quite considerably [15,16]. Injury of nerves in the lower pelvic floor is associated with surgical intervention to the prostatic gland in men, but it also occurs in mothers after complex delivery. Especially forceps-aided deliveries inherit a significant risk of damaging the sphincter complex [17]. Accordingly, in animal models, dual injury was applied to study insufficiency of the urethral closure apparatus [18]. To this end, pudendal nerve damage enforced by nerval crush—and urethral stretch—simulated by vaginal balloon dilatation—were performed as single and combined intervention [18]. While balloon dilatation injures the complete length of the urethra, local injury may also contribute to SUI symptoms. This was investigated by applying heat as used for coagulation of blood vessels during surgery. The so-called periurethral electrocautery of the middle stretch of the rat urethra caused a transient reduction of the leak point pressure. Histological analyses suggested that the effect noticed was a combined but locally defined tissue destruction of the rhabdosphincter muscle and the nerval connection [19].

In rats, a long-lasting urinary incontinence was also established by surgical removal of parts of the sphincter muscle, called urethrolysis [20]. This produced SUI-like symptoms for about 6 months. A different approach in rats was utilized when the pubo-urethral ligament was surgically transected. Here, in the short-term (approx. 1 week) and mid-term (approx. 4 weeks) leak point pressure was reduced [21,22]. A rather recent approach utilized old female rats expressing a transgene to cause severe obesity [23]. Transgenic rats presented increased voiding frequency, reduced leak point pressure and histological abnormalities in the sphincter muscle tissue. This suggested that obesity facilitates urinary incontinence by atrophy and possibly by distortion of urethral muscles [23]. In humans, obesity, including visceral intraabdominal fat affecting the pelvic floor vs. extra-abdominal adipose tissue with minor direct effects on the pelvic muscles, is sometimes noted in individuals with little physical activities and a not age-appropriate life style. This may facilitate a general decline of tissue qualities, including possibly sphincter deficiencies.

For experimental induction of urinary incontinence female animals were employed in most studies. However, in old male rats incontinence was induced by electrocautery of the rhabdosphincter. Electrocautery caused an irreversible destruction of the urethral muscle and nerve endings [24]. Transplantation of muscle sheets or muscle-derived cells regenerated sphincter function [24]. In male rabbits, urinary incontinence was induced by urethrolysis of the sphincter complex [25]. By cystometry and leak-point pressure measurements, incontinence and spontaneous functional regeneration were determined. Loss of continence was associated with a significant decrease in smooth muscle tissue and increased collagenous fibrosis [25]. To the best of our knowledge, this latter model was not employed in cell therapy studies so far.

Canine models of SUI were established as well using nerval injury or urethrolysis [26,27,28]. Urethral insufficiency was observed for approximately 2 to 7 months during follow up. In cats, SUI was induced by unilateral and bilateral neural lesions. Upon bilateral nerval injury, permanent urine leakage and significant reduction of urethral pressure were recorded [29]. Despite its clear outcome, this feline SUI model was not included in experimental therapy studies to the best of our knowledge.

Others used pigs to establish an SUI model by either urethral dilatation, or by urethral dilatation in combination with transurethral electrocautery of the closure complex [30,31]. In these animals, spontaneous regeneration of the sphincter deficiency was not observed for at least 3–4 weeks. A very interesting SUI model utilized aged multipari goats [32]. This model omits an experimental induction of sphincter insufficiency but relies on urethral dysfunction induced by mechanical stress of many deliveries and on the loss of tissue elasticity and muscular strength associated with aging [32]. With all limitations, an animal model comprises, this latter model probably reflects the pathology of SUI in postmenopausal multipari mothers better than all other quadruped animal models studied so far. In contrast, surgical induction of sphincter deficiency by injury of muscle tissue or peripheral nerves may come closer to the SUI of men after prostate surgery (Table 1).

### 2.2. Determining Incontinence in Animals 

Stress urinary incontinence is defined by the International Continence Society (ICS) as “the complaint of any involuntary loss of urine on effort or physical exertion or on sneezing or coughing” [33]. Patients can report urges to urinate, animals cannot. In patients, the upright posture facilitates involuntary loss of urine, while quadruped animals sometimes fail to spontaneously discharge from urine even after surgically established sphincter injury [34]. Moreover, dogs, cats and possibly a few other animals can be trained to avoid spontaneous micturition. In some of such “continent” pets, incontinence was observed when they were old and/or upon ovariectomy. However, such models of SUI seem less well predictable, not clearly defined nor easily manageable for investigating therapy of SUI. To examine continence versus incontinence in animals, experimental surrogates and substitute measurements must be employed in most circumstances. This includes the “sneeze testing” of animals and monitoring of the discharge, the tilt test of rodents, by which animals are fixed on a posture and tilted upright to determine loss of urine over time or determination of the urinary leak point pressure upon defined mechanical load [1]. However, such measurements on animals may be inconsistent and depend on the species and SUI model employed. To compensate for this, large cohorts yield robust results. Here rodents and other small animal models are advantageous. Last but not least, protocols should be standardized as much as possible in all animal studies. This includes but is not limited to (1) age, size, weight of the animals; (2) medication, anesthesia and positioning; (3) surgical procedures; (4) determination of urodynamics; (5) normalization of data measured.

Moreover, urethral continence is maintained by active and passive components, including smooth muscles of the lissosphincter as well as striated muscles of the rhabdosphincter [35]. Urinary continence is therefore partially under somatic neuronal control. When determining incontinence by measuring the urethral wall pressure in animals in sedation versus deep anesthesia, significant differences were noted disclosing the contribution of the neuronal control and striated muscles to the overall closure pressure [36]. The contribution of passive forces such as tissue tension, blood pressure to the overall continence is noticed, when urodynamics are measured in animals in anesthesia and upon sacrifice (unpublished observation). Still, determination of the leak point pressure and urethral wall pressure were the preferred techniques to determine sphincter insufficiency in large SUI models [30,31,32].

### 2.3. Exploring Cellular Therapies for Stress Urinary Incontinence in Large Animal Models

Female dogs also served as pre-clinical model to investigate regeneration of surgically induced SUI [2,28]. In one of these studies, the sphincter muscle was injured surgically to establish a long-lasting insufficiency. By measuring the leak point pressure in sham-treated versus myoblast-treated female dogs, sphincter regeneration was monitored. The best results were achieved when myoblast therapy was combined with electrophysiological stimulation [28]. This suggested that either the integration of myoblasts in the sphincter complex and their differentiation to become myofibers, the nerval connection, or both were improved by electrostimulation treatment [28]. Possibly also blood circulation and neovascularization were improved by electrostimulation.

Others used female landrace pigs to induce sphincter deficiency by dilatation of the urethra [30]. A significant reduction of the urethral closure pressure and of the functional length were determined after 4 weeks. The same group extended these studies and injected porcine muscle-derived cells in pigs pre-treated with sphincter insufficiency. In sharp contrast to other studies [28], cell therapy with muscle derived cells only did not yield functional recovery from SUI in these pigs [37]. This means that the outcome of both, experimentally induced incontinence, as well as the therapeutic benefit, depending on the individual design of the study. Nevertheless, using older multipari female goats as a SUI therapy model, a combination therapy of muscle derived cells and bone marrow-derived mesenchymal stromal cells yielded better regeneration when compared to mono therapies [32].

When investigating the efficacy of cell therapies of SUI and other diseases, the detection/localization of the cells applied, their viability and regenerative action are a concern. In many pre-clinical studies, cells were labeled by lipophilic, cell membrane-anchored fluorescent dyes such as PKH26 or alike [32,37,38]. One advantage of this type of label is the simple and straight forward method of labeling prior to the application intended: incubate the cells with the respective components as required, remove the dye and wash the cells. A suitable biosafety laboratory and a gene technology license are not required. This class of dyes comes in different colors and it is characterized by brilliant excitation-emission characteristics. The extinction coefficient *ε* of DIL and other lipophilic carbocyanides is very high and reaches more than 125,000 cm^−1^M^−1^ at the absorption maximum. Therefore, labeled cells can be detected easily with different visualization techniques in organs, tissue pieces or cryo- and microsections [39]. Moreover, by using complementary fluorescence labeled antibodies, detection of cells in one fluorescent channel can be combined with detection of several antigens by immunofluorescence in other channels, and/or with detection of cell nuclei stained e.g., by one of the Hoechst dyes, DAPI or alike. A general disadvantage of membrane anchored fluorescent labels is the fact that they are diluted with each cell division and therefore the label intensity may cease over time [31]. Of note, when labeling non-dividing cells or erythrocytes, labeled cells can be detected in vivo/ex vivo after several weeks of cell application. However, transfer of membrane anchored labels to neighboring cells or extracellular matrix components was reported. This transfer may occur from living cells and even more so from dead cells, including apoptotic cells or apoptotic bodies [40]. Therefore, such labels are not highly specific for the cells originally marked after in vivo application over time. Nevertheless, our recent studies indicated that the viability, proliferation, attachment and differentiation capacities of mesenchymal stromal cells were not modulated by the membrane-anchored fluorescent dyes PKH26 or VybrantDil. However, migration of the cells was reduced by loading the cell membranes by these labels [41].

Others utilized the expression of recombinant labels to visibly mark cells. To this end, the cells were transduced or transfected with viral, DNA or RNA vectors expressing for instance an enhanced green fluorescent protein (eGFP) [42]. By recombinant techniques, the fluorescent protein expressed can be specifically directed to sub-cellular compartments facilitating a localized coloring of cells predominantly in the nuclei, mitochondria, or other compartments. However, eGFP and other fluorescent proteins display extinction coefficients *ε* in the range of 55,000 cm^−1^M^−1^ at their absorption maximum, which is considerably lower when compared to typical lipophilic labels. Moreover, cell membranes provide a large area that can be charged with different intensities by lipophilic labels without interfering with physiological processes [41]. In contrast, intracellular structures such as nuclei or mitochondria are comparably small and cannot possibly be charged to the same extent with fluorescent proteins without influencing their physiological function. Therefore, recombinant labels grant lower fluorescence intensities in most experimental settings. However, this may be compensated by the detection of the recombinant protein by immune fluorescence. [43]. Recombinant fluorescent labels inherit other significant advantages. As mentioned above, subcellular structures can be addressed depending on the experimental need and their expression can be utilized as an indicator for an active metabolism and viability of the cells under investigation. By use of reporter constructs in combination with fluorescent protein expression in cells under exploration, modulation of gene expression can be monitored in vivo. The application of recombinant fluorescent labeled cells expands to the adoptive transfer of cells, tissues and even organs from transgenic, knock-in, knock-out or CRISPR-Cas-modified animals to e.g., wildtype animals. However, this large field of cell applications is beyond the focus of this review.

Bleaching by intensive UV exposure is always a concern with organic fluorescent dyes. Fluorescent nanoparticles (FNPs) are rather insensitive to bleaching. They also come with a wide spectrum of emission spectra, and—most important—a choice of size and shape, a feature not provided by other fluorescent labels. Nowadays, in different fields of biomedical research FNPs, so-called quantum dots (QDs) replace more and more the classical organic fluorescent labels [44,45]. Molecular interactions of proteins with proteins, proteins with nucleic acids, etc., or biosensing can be monitored by combinations of suitable QDs enabling the transmission of excitation energy between adjacent nanoparticles by fluorescence resonance energy transfer (FRET). However, FNPs can be picked up by cells through endocytosis, larger particles (1000 nm to 500 nm) rather than by phagocytosis, smaller ones by pinocytosis [46]. Therefore, in contrast to the recombinant labels described above, QDs may spread in vivo and over time to neighboring cells and structures, thus adding a level of complexity when critically evaluating the cell injection experiment. Additionally, QDs are also diluted with each cell cycle comparably to lipophilic membrane dyes. Moreover, labeling of cells by QDs may require a series of preliminary experiments to yield satisfactory results. In contrast, cell labeling by lipophilic days is a rather simple technology and optimized commercial kits are available for many and very diverse applications. The large field of applications of QDs in biomedical research merits a specific review. Hence, an in-depth discussion of QD labeling is beyond the horizon of this article. Some differences between organic fluorescent dyes and QDs are summarized in Table 2. In the end, the experimental design and aim, including especially the duration of follow-up of animal studies, have an impact on the labeling strategy. Dilution of labels as seen with QDs and lipophilic dyes is less critical for short-term follow-up and grants, in most circumstances, bright fluorescent signals. Long-term cell therapies may require the application of a “genetic label”, e.g., cells from suitable GFP-transgenic donors.

### 2.4. Porcine Animal Models in SUI Research

In our studies female landrace hybrid pigs (fLHP) as well as female Göttingen minipigs (fGMP) were included [31,34,38,43]. In a small feasibility study using cohorts of 2 or 3 female pigs each, we explored the muscular strength of the urethra of young virgin fLHPs (approx. 3 months of age, ±35 kg weight), versus young virgin fGMPs (approx. 9 months of age, ±25 kg weight), versus aged and obese virgin fGMPs (approx. 30 months of age, ±80 kg weight), versus retired multiparous breeder fGMP (approx. 24 months of age, ±50 kg weight). The muscular strength of the urethral sphincter complex was determined by measuring the urethral wall pressure. To this end, a catheter equipped with pressure sensors is introduced through the urethra in the bladder and slowly retracted while recording the pressure levels. This method is referred to as urodynamics (Figure 1). The functional urethral length determines the part of the urethra with a local pressure above the intravesical pressure in rest. Urethral closure pressure results from smooth muscle cells with autonomic innervation of the urethra and striated voluntary innervated muscle cells of the rhabdosphincter. Thereby, the maximum urethral closure pressure reflects the position of the most condensed accumulation of primarily striated muscle cells [47]. In addition, the area under the curve (AUC) reflects the definite integral of the urethral closure pressure curve, which is currently more interesting from a scientific than a clinical point of view (Figure 1). When comparing the urodynamics of young fLHPs versus size-matched young fGMPs, young versus old and obese virgin fGMPs, or old obese virgin versus normal multiparous retired breeder fGMPs, differences in urodynamics were noted (Figure 2). Interestingly, in contrast to obese rats [23], in our study old obese fGMPs did not show a reduced pclo nor reduced pmax (Figure 2). Histological analyses of the corresponding tissue samples are ongoing. However, the preliminary urodynamic data provide evidence that not only the species by itself, but also differences in the individual breed of a given species (here landrace hybrid versus Göttingen minipig), age, body mass, and breeder status may influence the outcome of functional studies of the urethra, especially in the context of diagnostic and therapeutic models of incontinence.

### 2.5. Porcine Animal Models in SUI Cell Therapy

Using the tools described above, we explored the potential of cell therapies for SUI in female pigs. In one series of experiments, we investigated the injection of human bone marrow-derived mesenchymal stromal cells in fGMP. These animals received immunosuppression by cyclosporine A (15 mg/kg body mass per day orally). Starting at about 9 months of age, the body mass of fGMP increased during follow-up of up the 12 months from 15–20 kg to 25–37 kg. Thus, the fGMP can be handled without difficulties even after extended incubation periods. Immediately after injection of 4 aliquots of 250 μL medium each, the maximum urethral closure pressure (pmax) was significantly reduced from 142.4 ± 31 cm H_2_O to 116 ± 38 cm H_2_O (*p* < 0.09; Figure 3). The AUC (*p* < 0.05; Figure 4) and the functional length (not significant, not shown) was reduced as well.

This indicated that injection of even small volumes of isotonic solvent caused a significant irritation of the closure complex in this large animal model. Swelling or bulking effects were not observed. We hypothesize that cell injection in smaller animals may be even more detrimental. However, no significant differences in pmax were recorded in sham-treated animals during follow-up for up to 12 months in fGMPs (Figure 5).

While the functional urethral length did not show any changes between sham animals during the follow-up (Figure 5), significant differences in AUC were observed between the sham-treated cohort when compared to animals after 6 months of follow-up (*p* < 0.02; Figure 6). We conclude that even minor injuries may modulate the function of the porcine sphincter. 

The data suggested that needle injections of small aliquots of isotonic fluid caused a significant short-term reduction of sphincter function but no long-term bulking. Our histological analyses did not indicate that inflammation or fibrosis occurred [38].

In the next series of experiments, we injected human placenta-derived stromal cells in healthy, i.e., not incontinent fGMPs. As seen in sham-treated fGMPs (Figure 5), three weeks after injection a minor but significant slope of the maximal urethral wall pressure was recorded (ANOVA: *p* < 0.04; Figure 7), followed by a transient but significant increase of pmax (ANOVA: *p* < 0.03; Figure 7). However, upon Tukey–Kramer analysis only the difference between 3 weeks and 3 months follow-up were significant (*p* < 0.024, Figure 7). Significant differences in AUC and in functional lengths of the urethra were also observed between controls and injected animals for some of the timepoints investigated (not shown).

In another series of experiments, porcine adipose-derived stromal cells [43] or muscle-derived cells (unpublished results) were injected in fLHPs. In such pigs, induction of a transient urethral insufficiency was generated by urethral dilatation and electrocautery about 1–2 cm distal of the bladder neck [31]. This corroborated other studies [30]. Upon combining urethral dilatation with electrocautery, urethral insufficiency lasted at least three weeks [31]. Current studies are designed to transfer this method of induction of experimental SUI from fLHPs to fGMP to facilitate long term follow-up after cell therapy. We recently noted that the differences in anatomy and probably physiology of bladder and urinary tract may yield differences in the functional lengths, urethral wall pressure profiles and other physiological parameters in different breeds of the same species (Figure 2). Moreover, preliminary data also suggest that not only the breed but also age and multiple deliveries have an impact on urodynamic parameters in different porcine breeds (Figure 2). This to the best of our knowledge is not investigated in detail yet, but may of course influence the outcome of cell therapy studies in both, experimentally or naturally incontinent animals after SUI therapy in comparison to different control cohorts included in such pre-clinical studies.

## 3. Conclusions

In preclinical research, large animal models inherit several disadvantages and require larger efforts to yield robust and statistically reliable results. Moreover, biological tools such as specific antibodies, nucleic acid probes or alike and bioinformatic data are available on a large scale for studies employing human and e.g., rodent samples, but are accessible to a much lesser extent for studies with large farm animals including goats, sheep, or pigs. However, on the other hand, in SUI research, larger species grant several advantages including transurethral surgery with standard instruments. This opens different possibilities to develop novel or improved surgical instruments in pre-clinical studies, basically at a 1:1 scale. Therefore, for preclinical studies focusing on surgery or medical technology, we highly recommend farm animal models. Studies dealing with SUI and physiology or anatomy might require primate models of incontinence.

## Figures and Tables

**Figure 1 ijms-22-06092-f001:**
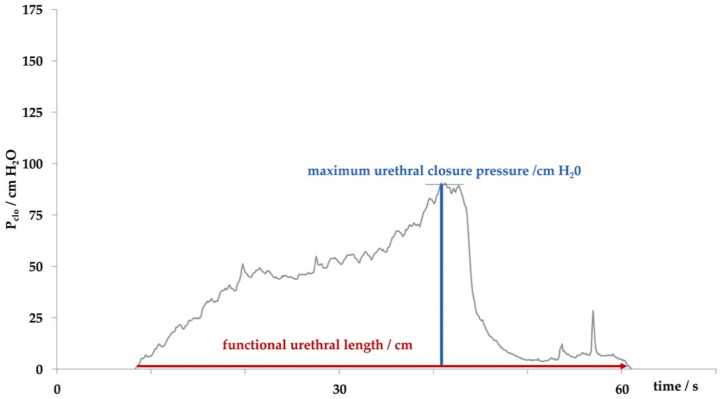
Determining the sphincter function by urodynamics. The functional urethral length determines the part of the urethra with a closure pressure above the intravesical pressure. The maximum urethral closure pressure describes the location of accumulation of sphincter muscle cells, which is mostly due to striated cells of the rhabdosphincter. The area under the curve (AUC), the area described by the grey histogram and x-axis, determines the integral of urethral wall pressures of the functional urethral length.

**Figure 2 ijms-22-06092-f002:**
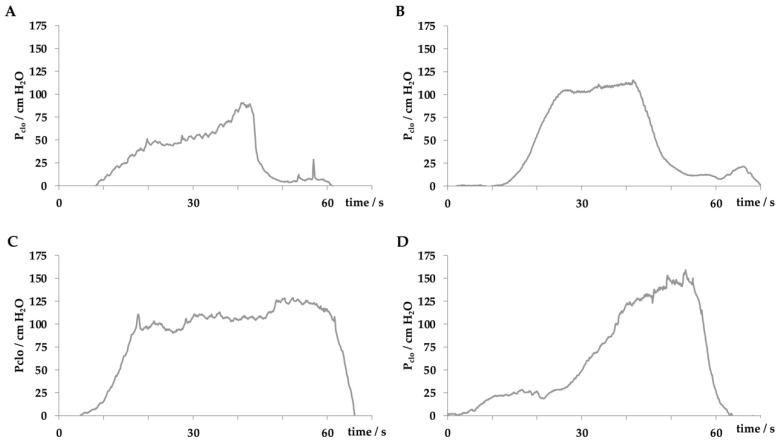
Comparing the urodynamic profiles in female pigs including young virgin fLHP (**A**), young virgin fGMP (**B**), old obese virgin fGMP (**C**), and retired breeder fGMP (**D**).

**Figure 3 ijms-22-06092-f003:**
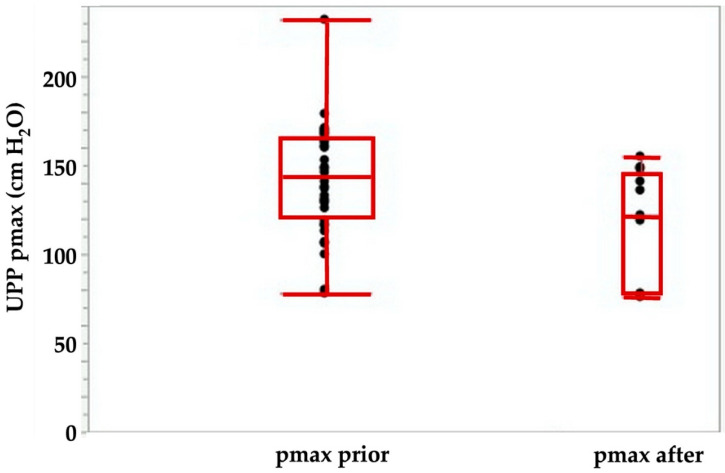
Maximal urodymanic closure pressure pmax determined by urodynamic pressure profilometry (UPP) before (left) versus immediately after injection of 4 aliquots of 250 μL medium. The pmax determines the maximal muscular sphincter force and its localization in the urethra.

**Figure 4 ijms-22-06092-f004:**
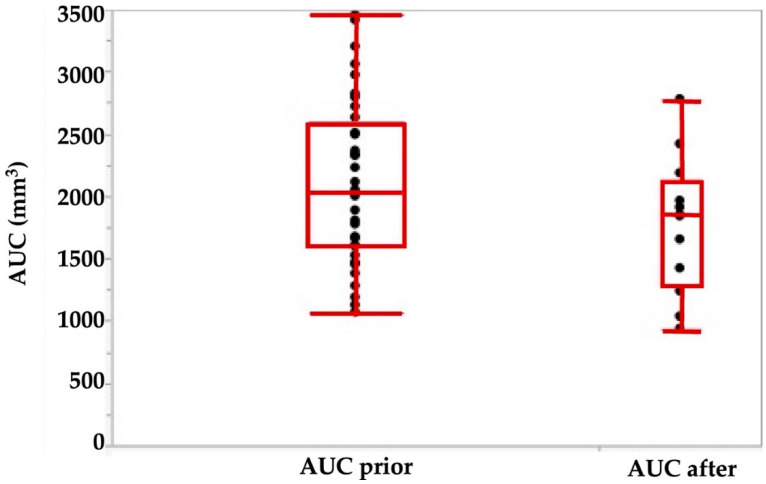
Area under the curve (AUC) determined by urodynamics before (left) versus immediately after injection of 4 aliquots of cells in 250 μL medium each. The AUC is calculated as the integral of the total urethral wall pressure over the total urethral length.

**Figure 5 ijms-22-06092-f005:**
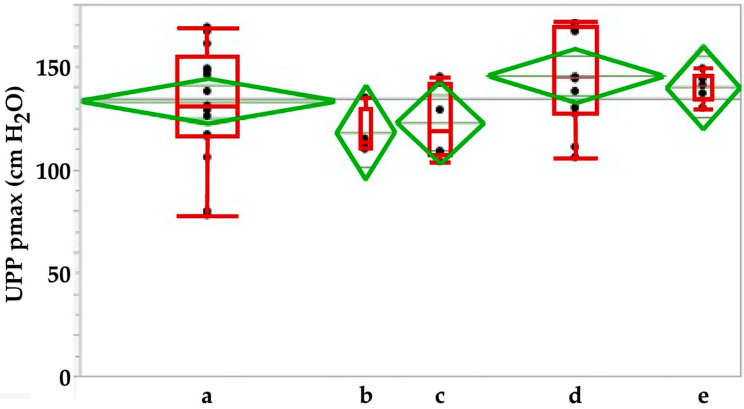
Maximal urodymanic closure pressure pmax in sham-treated animals immediately after sham-surgery (**a**) in comparison to animals during follow-up for 3 weeks (**b**), 3 (**c**), 6 (**d**) or 12 (**e**) months after sham treatment. The pmax determines the maximal muscular sphincter force and its localization in the urethra.

**Figure 6 ijms-22-06092-f006:**
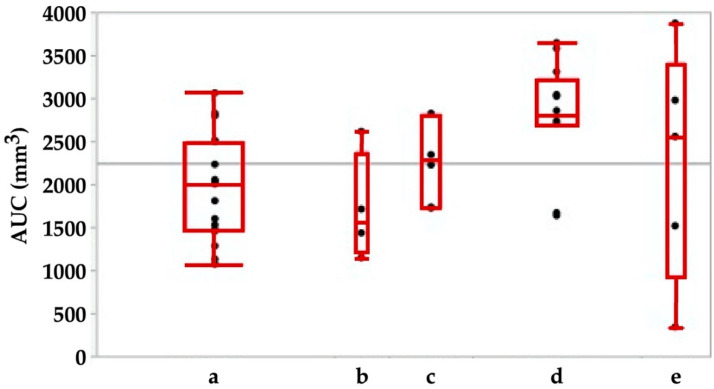
Area under the curve (AUC) in animals immediately after sham-treatment (**a**) in comparison to animals during follow-up for 3 weeks (**b**), 3 (**c**), 6 (**d**) or 12 (**e**) months. The AUC is calculated as the integral of the total urethral wall pressure over the total urethral length.

**Figure 7 ijms-22-06092-f007:**
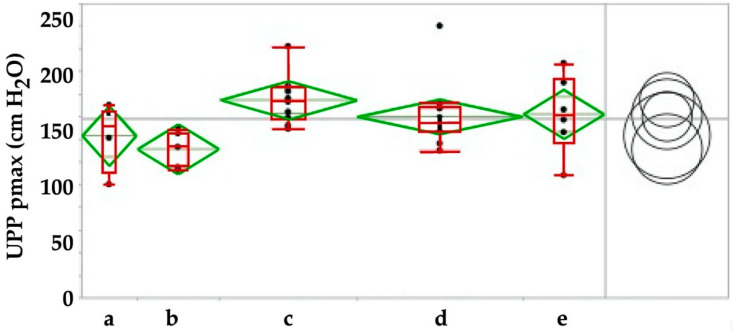
Maximal urodynamic wall pressure before (**a**) versus injection of human placenta-derived mesenchymal stromal cells and during follow-up over 3 weeks (**b**), 3 (**c**), 6 (**d**) and 12 (**e**) moths respectively. By Tukey–Kramer test only the increase in pmax between f/u of 3 weeks versus 3 months was significant, pmax at all other timepoints were not significant (right panel). The pmax determines the maximal muscular sphincter force and its localization in the urethra.

**Table 1 ijms-22-06092-t001:** Animal models to investigate etiology, diagnosis and therapy of SUI.

Induced Models		
	Transient/Short Term	Permanent/Long Term
	urethral dilatation	urethrolysis
	local electrocautery	pubourethral ligament transsection
	vaginal distension	pudendal nerve transsection
	pudendal nerve crush	bilateral nerve resection
		enforced electrocautery
		urethral sphincterectomy
**Spontaneous Models**		
		multipari old female animals
**Genetic Models**		
		transgenic animals
		knock-out animals

**Table 2 ijms-22-06092-t002:** Some basic characteristics of organic fuorescent dyes compared to quantum dots.

Properties	Organic Dye	Quantum Dot
Molecular absorption coefficient *ε*	moderate: 2.5 × 10^4^ to 2.5 × 10^5^ M^−1^cm^−1^	high: 1 × 10^5^ to 1 × 10^6^ M^−1^cm^−1^
Emmission spectra, width	variable, 30 nm to 100 nm	variable, 30 nm to 90 nm
Emmission histogram	asymetric, often tailing to long wavelength	symmetric, Gaussian profil
Fluorescence duration	short: 1–10 ns	longer: 10–100 ns
Photochemical stability/bleaching	sufficient/depending on UV input	very high, photo-brightning possible
Toxicity to cells	very variable, depending in part on solvent	variable, depending on structure/chemistry
Signal amplification	possible, e.g., by immune fluorescence	often not needed due to bright/stable signals
Labeling method	very simple, commercial kits available frommany providers for almost any application	may require specific protocols for each application, may need functional groups on target molecules for interaction

## Data Availability

Not applicable.

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
