# Peer review of "Large Animal Models for Investigating Cell Therapies of Stress Urinary Incontinence"

_ijms, 2021, doi:10.3390/ijms22116092_

Round 1
Reviewer 1 Report
I read with a really great interests paper entitled “Large Animal Models for Investigating Cell
Therapies of Stress Urinary Incontinence”. The authors did a very interesting narrative review for SUI among animals and potential methods of therapy.
I have only one minor comment:
It would be nice to explain each abbreviation in description of each figure (i.e. UPP, AUC).
I would like to recommend this paper for publication.
Author Response
Comments to reviewer 1:
We thank this reviewer for the comments and suggestions and revised the manuscript accordingly: We complemented the legends to Figures 1, 3, 4, 5, 6, 7 to better explain the data presented to the readers not familiar with Urology.
We also added the abbreviation UPP and a definition this term to the list of abbreviations (lines 390-391). UPP was used e.g. in figure 3.
All changes dedicated to this review are highlighted by red letters
Reviewer 2 Report
This manuscript reviewed recently published large animal researches on stress urinary incontinence. The authors have thoroughly described the advantages and disadvantages of experimental techniques especially in using cell therapy.
The review is extensive but might not suitable for this molecular journal.
Author Response
Comments to reviewer 2:
We thank reviewer 2 for the comment and respond to the concern brought forward as follows:
This review is part of a Special Issue dealing with cell based therapy of stress urinary incontinence. We agree that the Special Issue, to which this manuscript was submitted, is not dealing specifically and only with research in molecular scales or terms. In contrast, this Special Issue is dedicated to a clinical challenge - urinary incontinence - which at present can't be treated in many cases sufficiently. Therefore novel strategies including cellular and molecular active components will be required.
Developing novel regimen for clinical use requires intensive research in the lab and in suitable animal models to gain data on safety and efficacy of novel drugs. Some of them appear nowadays on the horizon: For instance, strategies employing iPSC-derived regenerative cells are under investigation. Extracellular vesicles including exosomes are in the focus of experimental studies.
Our animal studies are on the way to bridge the current gap between the mostly conservative and surgical regimen used nowadays and very promising strategies employing e.g., autologous iPSC-derived muscle cells, exosomes and other cellular or molecular active components.
This review therefore presents an important corner stone in development of future cellular and molecular therapies to better treat urinary incontinence. We therefore are convinced that it fits very well in a Special Issue dealing with a disease that affects many people world wide and urgently needs better therapies.
Reviewer 3 Report
The manuscript, "Large Animal Models for Investigating Cell Therapies of Stress Urinary Incontinence" is a nice and (as far as I know) complete narrative overview of all animal models to test stress urinary incontinence therapies. Sphincter muscle weakness and / or insufficient ability to remain closed with sufficient pressure to withstand the opening forces that occur with increasing pressure in the abdomen is the paradigm of stress urinary incontinence in humans. It is generally accepted that delivery through the vaginal canal, through the pelvic floor, causes more or less damage to the muscles, as well as damage to blood flow and / or innervation. It is also widely accepted that lack of muscle use or exercise leads to loss of strength and it is also believed that chronic pelvic floor muscle overload would impair function and / or strength. Furthermore it is accepted that the passage of time (aging) associates with decay of striated muscle quality. Most animal models are an attempt to mimic the pathophysiology described in the manuscript. Clinical testing for SUI is not complex; urine loss with exertion is SUI, and invasive urodynamic testing is useful in a proportion of patients to diagnose or exclude other dysfunction of the loser urinary tract. Urethral pressure profile has little value in the (clinical) diagnosis see (e.g. international consultation on incontinence reports, or other guidelines) but is the best manner to compare maximum urethral closure pressure and may be useful in clinical subgroups stratify for specific measures. Most of the animal experiments use urethral pressure or a derivative of it to evaluate changes after intervention. some use the observation of incontinence in association with abdominal pressure. I have few recommendations for the manuscript: First: In humans the pelvic floor is weight bearing in vertical position. The abdominal mass (from diaphragm to below) lies on the pelvic floor. This is not the case in quadrupeds. This is –in my view- a potential bias/shortcoming in models that aim to mimic chronic strain and or ageing. Furthermore, in association with this: obesity adds mass, but most of it outside the abdominal cavity. Most of the gained weight is not sitting on the pelvic floor, and –in my view- even less weight –incrementing in quadrupeds. I think this should be considered (explained in the manuscript) for the presented models. But: obesity may combine with physical inactivity and via that with generalized decline of muscle quality…. Another comment: Humans can voluntary inhibit and initiate voiding, and use this for socially (in evolution: biologically-) acceptable voiding. Some animals do void in specific places and are able to inhibit, but some not. But also, some (many small) animals do no void synergic and or complete (uninterrupted). Although this is not SUI (and mentioned in the manuscript) I think that this may deserve some more specific explanation (in the manuscript) of the relevance and physiology of spincteric (closure) function in the mentioned animals. Especially also L127 ‘uncontrolled loss of urine’ should be (better) specified (in this perspective) (I am not sure whether animals can have DO and or DO-incontinence (they can have increase in voiding frequency, indeed), but this is –acceptable- not the scope of this manuscript). One comment about the (repeated) urethral pressures. I think that it is relevant to include whether the tests (repeated/comparative) were done in the identical position of the animal (and or corrected for or measured relative to intravesical pressure). One other: I appreciate the word ‘urodymanic’ in the legend of fig 7, however I assume that this is a typing error. Last but not least, I think that the conclusion can be much more specific about what this manuscript has summarized. The conclusion is now very similar to the introduction/aim of the manuscript.
Author Response
Comments to reviewer 3:
We thank reviewer 3 for the detailed and very informative comments and suggestions to our review. we have addressed all of them and marked the changes in the revised manuscript by blue letters.
Specifically:
- The reviewer raised an interesting point regarding the anatomy of quadruped animals. This statement was complemented by the fact that in animals less weight is pressing on the bladder in the lower pelvic floor (lines 36, 37 in the revised manuscript).
- We thank for the comment on the possible link between obesity and incontinence in human patients. We complemented the revision in lines 98 - 101 accordingly.
- The third comment points at voiding habits of different animal species. Besides discharge, many animals use voiding to e.g., mark their territories, to communicate within a herd, or to inform about heat/runt etc. To the best of our knowledge, these aspects have not been included at large in SUI studies and literature dealing with preclinical animal SUI studies. However, in rodent studies, animals voided prior to experimental SUI-induction in a latrine corner of the cages. This was documented. After SUI-induction, rats voided statistically everywhere in the cages. This was documented and used to prove incontinence. But this change in voiding seems not to relate to the physiology of the (quote) "sphincteric closure function" but is part of the induced SUI model. Due to the length of the current manuscript we would prefer not to add and discuss this aspect.
- We appreciate this point of critique as the phrase (quote) "uncontrolled loss of urine" is neither defined in the review nor used by the definitions coined by professional societies including the ICS. We therefore rephrase this section accordingly (see line 133 of the revised manuscript).
- The need for standardized study design including e.g., animal position, correction of UPP to intravesical pressure etc. is key to the quality of results obtained from animal studies. We agree to this comment of reviewer 3 and complemented the review accordingly (see lines 144 - 147 in the revised manuscript).
- We apologize for the typographical error in the legend to figure 7 and corrected it.
- We agree to this comment but in a review, there is no classical conclusion from new data reported in comparison to the state-of-the-art. We therefore only moderately revised the conclusions (see lines 359- 368 in the revised version of the manuscript).
Round 2
Reviewer 2 Report
No further comment.